# Predictive Factors for Critical Weight Loss in Saudi Head and Neck Cancer Patients Undergoing (Chemo)Radiotherapy

**DOI:** 10.3390/cancers16020414

**Published:** 2024-01-18

**Authors:** Alaa S. Redwan, Fayrooz A. Kattan, Maha A. Alidrisi, Gayur A. Ali, Mazen M. Ghaith, Ahmad F. Arbaeen, Hussain A. Almasmoum, Najlaa H. Almohmadi, Sarah O. Alkholy, Walaa E. Alhassani, Wafaa F. Abusudah, Abrar M. Babateen, Mai A. Ghabashi, Ahmad A. Obeidat, Azzam N. Al Yacoub, Awfa Y. Alazzeh, Firas S. Azzeh

**Affiliations:** 1Clinical Nutrition Department, Faculty of Applied Medical Sciences, Umm Al-Qura University, P.O. Box 715, Makkah 21955, Saudi Arabia; redwan.a@kamc.med.sa (A.S.R.); nhmohmadi@uqu.edu.sa (N.H.A.); sokhouli@uqu.edu.sa (S.O.A.); wehassani@uqu.edu.sa (W.E.A.); wfabusudah@uqu.edu.sa (W.F.A.); ambabteen@uqu.edu.sa (A.M.B.); maghabashi@uqu.edu.sa (M.A.G.); 2Clinical Nutrition Administration, King Abdullah Medical City, P.O. Box 24246, Makkah 21955, Saudi Arabia; kattan.f@kamc.med.sa; 3Radiation Oncology Department, King Abdullah Medical City, P.O. Box 24246, Makkah 21955, Saudi Arabia; aledreesi.m@kamc.med.sa (M.A.A.); musharraf.g@kamc.med.sa (G.A.A.); 4Department of Clinical Laboratory Sciences, Faculty of Applied Medical Sciences, Umm Al-Qura University, P.O. Box 715, Makkah 21955, Saudi Arabia; mmghaith@uqu.edu.sa (M.M.G.); afarbaeen@uqu.edu.sa (A.F.A.); haamasmoum@uqu.edu.sa (H.A.A.); 5Department of Nutrition and Food Technology, School of Agriculture, The University of Jordan, Amman 11942, Jordan; ahmad.obeidat@ju.edu.jo; 6Department of Biology, Faculty of Applied Science, Umm Al-Qura University, P.O. Box 715, Makkah 24382, Saudi Arabia; anyacoub@uqu.edu.sa; 7Department of Clinical Nutrition, Faculty of Applied Medical Sciences, University of Ha’il, Ha’il 21955, Saudi Arabia; a.alazzeh@uoh.edu.sa

**Keywords:** head and neck cancer, critical weight loss, Makkah, radiotherapy, chemotherapy

## Abstract

**Simple Summary:**

Patients with head and neck cancer (HNC) often experience weight loss due to the tumor and its treatment. This article aims to investigate the possible factors that contribute to critical weight loss (CWL) in patients with HNC who have undergone radiotherapy or concurrent chemoradiotherapy. Based on this study, patients with HNC who are undergoing radiotherapy with or without chemotherapy and have a BMI equal to or greater than 25 kg/m^2^ should be closely monitored in terms of nutritional assessment, intervention, and toxic side effects. Implementing these practices is expected to improve treatment outcomes for HNC patients and close the gap between researchers and policymakers. The study also encourages further research to better understand the relationship between overweight and obese HNC patients and CWL.

**Abstract:**

Weight loss is a significant health problem among patients with head and neck cancer (HNC) that is attributable primarily to the tumor or tumor therapy. Critical weight loss (CWL) is defined as the unintentional loss of ≥5% of weight. Therefore, this study’s goal was to investigate and determine the possible factors influencing CWL among patients with HNC who have received radiotherapy or concurrent chemoradiotherapy (CCRT). We conducted a retrospective analysis of 175 patients who received radiotherapy or CCRT as either their primary, adjuvant, or combined treatment at the Oncology Center in King Abdullah Medical City. All patients were ≥18 years of age and diagnosed with HNC with no metastasis. The study results showed that 107 patients (61%) had CWL, while 68 (39%) did not. The following factors were significantly predictive of CWL with a multivariate regression analysis: pretreatment BMI (AOR = 1.1, 95% CI = 1.02–1.17), oral cavity cancer (AOR = 10.36, 95% CI = 1.13–94.55), and male sex (AOR = 3.15, 95% CI = 1.39–7.11). In conclusion, weight loss is highly prevalent among HNC patients during treatment. Accordingly, pretreatment BMI, cancer in the oral cavity, and being male can be considered predictive factors for CWL.

## 1. Introduction

Cancer is the second leading cause of death worldwide and accounts for approximately 10 million deaths annually [1]. According to the Global Burden of Disease study, 890,000 new head and neck cancers (HNCs) (lip and oral cavity, nasopharynx, pharynx, and larynx] were diagnosed globally in 2017, which constituted 5.3% of all cancers (excluding nonmelanoma skin cancers) [2]. HNCs have a 5-year survival rate of approximately 50%, which has remained constant in recent years [3]. In Saudi Arabia, there were 24,485 new cancer cases and 10,518 cancer deaths in 2018 alone [1], and the prevalence of oral cancer in southern Saudi Arabia was 3.29% over the past ten years [4]. Some geographic differences in prevalence are linked to various risk factors that are distinct in different parts of the world [4]. Tobacco and alcohol, among the many etiological factors linked to cancer, have long been identified as known risk factors for oral cancer in individuals of all ages, and tobacco use in smoking cigarettes is universal around the world. An additional significant contributor to HNCs’ higher occurrence in Yemen and the southern region of Saudi Arabia is the use of *shemma* [4].

Critical weight loss (CWL), defined as the unintentional loss of weight ≥5% during radiation and CCRT, ranged from 37% in a mixed group of HNC patients to 88% in nasopharyngeal cancer patients and is related to acute toxicity of radiation and concurrent chemotherapy [5]. Furthermore, as a known negative prognostic factor, CWL increases the death rate in head and neck cancer patients. In previous studies, patients with a CWL of 5.4% attributable to the toxicity radiation causes had a poor prognosis [6]. In this cancer population, cumulative weight losses have been shown to surpass 20% of pretreatment body weight. A greater degree of weight loss has been linked to higher mortality and morbidity, treatment delays, longer hospital stays, and poorer performance status and quality of life [7]. In particular, a study found that a low pretreatment body mass index (BMI) and greater weight loss during treatment are associated independently and markedly with poorer survival, even when other established factors are considered [8].

Although numerous studies are currently underway in this area, no definitive conclusion has yet been established, as certain risk factors have not proven to be reliable indicators. We hypothesized that HNC patients are at high risk of weight loss during radiotherapy (RT) (+/− chemotherapy), which is considered, with BMI as the pretreatment, a prognostic factor. Weight loss is a significant health problem among patients with HNCs that is attributable primarily to the tumor or tumor therapy [9]. Therefore, this study’s primary goal was to investigate and determine the possible factors that influence CWL among patients with HNC who received radiation or concurrent chemoradiotherapy (CCRT). The study’s secondary goal was to demonstrate the potential toxicities’ correlation with the weight loss percentage.

## 2. Materials and Methods

### 2.1. Study Setting and Design

This study conducted a retrospective analysis of 175 patients who underwent RT or CCRT either as a primary, adjuvant, or combined treatment at the King Abdullah Medical City (KAMC) Oncology Center. Data were collected from February 2015 until August 2021 according to eligibility criteria. Patient information was collected from electronic medical records systems, Varian (specified for oncology patients undergoing RT), and TrackCare.

### 2.2. Patient Eligibility

The patients included were ≥18 years of age, diagnosed with HNC, intended for curative treatment of RT or CCRT, and had their weight (kg) and height (cm) documented before and after completing the treatment. The HNC patients excluded were those with distant metastases, receiving treatment with palliative intent, those who discontinued treatment or died during the course of therapy, and any patient with missing documentation of study variables. Patient TNM staging (T refers to the primary tumor, N refers to whether lymphatic nodes are affected, and M refers to further metastases) (National Cancer Institute; NIH, 2015) was classified into four groups according to cancer stage: I, II, III, and IV. Cancer stages were classified further into two groups: early (stages I and II) and advanced (stages III and IV) [10]. The sample distribution according to the inclusion and exclusion criteria is described in Figure 1. In addition, tumor site was categorized into four groups: cancer in the oral cavity, pharynx (oro-, naso-, or hypopharynx), larynx, and another group that included the neck and maxilla.

### 2.3. Data Collection

The data collected included demographic information, diagnosis and tumor site, surgical history, and TNM stage. Weight and height records were collected for all eligible patients, as documented by a nurse using a digital weighing scale with built-in height rods (Health O Meter, model: 597KL, 11800 S Austin Ave, Alsip, IL, USA) with a capacity of 272 kg and a ± 0.1 kg accuracy, both before the beginning of RT and at the end of the radiation course of treatment. Height and weight were recorded as described in the International Anthropometric Standardization reference manual [11].Participants were instructed to stand straight, wear light clothing, and be barefoot for all measurements. Using these data, we calculated the patients’ percentage of weight loss and further classified them according to weight loss percentage into CWL (≥5%) and non-CWL (<5%) groups [12]. The BMI categories were grouped according to the WHO classification of BMI (WHO, 2021): <18.5 kg/m^2^ (underweight), 18.5–24.9 kg/m^2^ (normal weight), ≥25–29.9 kg/m^2^ (overweight), and ≥30 kg/m^2^ (obese).

### 2.4. Treatment and Toxicities

All HNC patients included in the study were treated using RapidArc^TM^ (Varian Medical Systems, TrueBeam, Palo Alto, CA, USA) with a dosage grouping of ≥60 Gy (standard radiotherapy fraction dose) and <60 Gy. In the case of CCRT, chemotherapy was planned by an oncologist and administered to patients concomitantly, as appropriate. A radiation oncology specialist reviewed the toxicities assessment sheet in each patient’s file record on a weekly basis during the course of treatment. The weekly assessment sheet was approved at the KAMC Oncology Center and established according to the National Cancer Institute (NCI) Common Toxicity Criteria (NCI-CTC) scores chemotherapy (CT)-related side effects v. 2.0 [13], which has been used in all National Cancer Institute (NCI) clinical trials since March 1998 [14]. In this study, we collected the highest-grade severity reached and documented during radiation sessions for mucositis, dysphagia, mouth dryness, anorexia, and pain. 

### 2.5. Statistical Analysis

Data analysis was performed using SPSS v. 28 (IBM Corp., Armonk, NY, USA). All categorical variables are summarized as percentages and frequencies, while quantitative variables are reported as the mean ± standard deviation. Comparisons were made by performing the χ^2^ test or independent *t*-test for non-continuous and continuous parameters, respectively. The results are also presented as the odds ratio (OR) and adjusted odds ratio (AOR) using univariate and multivariate logistic regression tests, respectively, along with the inclusion of 95% confidence intervals (95% CI) to identify predictors of CWL in HNC patients. The significance level was set at *p* < 0.05.

## 3. Results

A total of 175 patients were included in the study (Table 1). Of these, 119 (68%) were men, and 56 (32%) were women. The patients’ mean age (years) was 54.6 ± 15.1, and their mean height (cm) was 163.6 ± 10.3. With respect to the pretreatment parameters, their mean pretreatment weight (kg) was 70.8 ± 19.5, and the mean pretreatment BMI (kg/m^2^) was 26.2 ± 6.2. Concerning the post-treatment parameters, their post-treatment weight (kg) was 65.7 ± 17.7 and their post-treatment BMI (kg/m^2^) was 24.4 ± 5.7, with a mean weight loss percentage of 6.7% ± 6%. Furthermore, 107 patients (61%) had CWL, while 68 (39%) did not. Regarding patient categorization based on BMI, we observed that 9% were classified as underweight, 37% had a normal BMI, 28% were overweight, and 26% were classified as obese.

The comparison between critical and non-critical weight loss among patients who had undergone radiotherapy for several variables is presented in Table 2. A significant association was found between sex and weight loss percentage (*p* = 0.005), in which 81 (76%) of men had CWL, while only 26 (24%) of women did. The diagnosis was also associated significantly with weight loss (*p* < 0.001), in that the highest rate of CWL was seen in patients with pharyngeal tumors (51%) and the lowest was seen in patients with neck and maxilla tumors (1%). Furthermore, there was a significant relationship among pharyngeal cancers according to the tumor site (oro-, hypo-, and nasopharynx) with CWL (*p* < 0.001). A significant relationship was found between weight loss percentage and disease stage (*p* = 0.02). Notably, 90% of patients in the advanced stages had CWL compared to only 10% of those in the early stages. History of surgery (tumor excision) was associated significantly with weight loss (*p* = 0.004), in which those without a history of surgery had a significantly higher rate of CWL than those who underwent surgery (78% vs. 22%). The intention of treatment was related significantly to weight loss (*p* = 0.001), in which those who underwent chemotherapy treatment had a significantly higher CWL rate than those without chemotherapy (77% vs. 23%). The radiation dose was related significantly to weight loss as well (*p* = 0.008), in which those with a radiation dose ≥60 Gy had a significantly higher rate of CWL than those with a radiation dose <60 Gy (98% vs. 2%). Both pretreatment weight and BMI were significantly higher in patients with CWL (*p* = 0.001 and 0.01, respectively).

Table 3 shows the regression analysis for the prediction of CWL in HNC patients during treatment. The factors included in the model were pretreatment BMI, sex, diagnosis, pharyngeal tumor site, disease stage, history of surgery, intention of treatment, and radiation dose. The following factors were significantly predictive of CWL with a multivariate regression analysis: pretreatment BMI (AOR = 1.1, 95% CI = [1.02–1.17], *p* = 0.007), oral cavity cancer (AOR = 10.36, 95% CI = [1.13–94.55], *p* = 0.04), and male sex (AOR = 3.15, 95% CI = [1.39–7.11], *p* = 0.006). In addition, other factors were found to be statistically significant with univariate logistic regression as a predictive factor, which included male sex (OR = 2.46, 95% CI = [1.28–4.71], *p* = 0.007), nasopharyngeal cancer (OR = 7.85, 95% CI = [3.26–18.88], *p* = 0.007), patients with advanced disease stages (stages III and IV) (OR = 2.68, 95% CI = [1.16–6.21], *p* = 0.02), treatment with CCRT (OR = 3.09, 95% CI = [1.6–5.94], *p* <0.001), and radiotherapy dose of ≥60 Gy (OR = 7, 95% CI = [1.43–34.04], *p* = 0.02). Additionally, tumors in the oral cavity, pharynx, and larynx showed risk factors for CWL (OR = 9.56, 95% CI = [1.14–79.79], *p* = 0.04, OR = 35.35, 95% CI = [4.12–302.83], *p* = 0.001, and OR = 11.76, 95% CI = [1.31–105], *p* = 0.03, respectively). On the other hand, protective factors included a history of surgery before radiotherapy treatment (OR = 0.38, 95% CI = [0.2–0.75], *p* = 0.005), and higher BMI (OR = 0.93, 95% CI = [0.88–0.98], *p* = 0.01). Upon conducting multivariate regression analysis, several variables that were previously found significant in the univariate analysis, such as tumors in the pharynx and larynx, nasopharyngeal cancer, advanced disease stage, history of surgery, treatment with CCRT, and radiotherapy dose of ≥60 Gy, were no longer significant. Interestingly, pretreatment BMI initially showed a protective effect but became a risk factor after applying the multivariate regression.

Figure 2 and Figure 3 illustrate the relationship between the incidence of mucositis, mouth dryness, and CWL. A significant relationship was found between developing mucositis and CWL at a rate of 58% vs. 32% (*n* = 39 vs. 22, respectively) in the non-CWL patients (OR = 3.6, 95% CI = [1.28–10.13], *p* = 0.02). Furthermore, the incidence of mouth dryness was higher in patients with CWL at a rate of 57% vs. 31% (*n* = 61 vs. 33, respectively) among patients with non-CWL (OR = 3.37, 95% CI = [1.27–8.96], *p* = 0.02). No significant difference was found in the other side-effects categories.

## 4. Discussion

HNC is prevalent in Saudi Arabia and worldwide. Weight loss and pretreatment BMI were significantly associated with HNC patients and may be considered prognostic factors. This study showed that high pretreatment BMI, cancer in the oral cavity, and being male were predictive factors for CWL. Univariate regression analysis showed that patients with nasopharyngeal cancer, being in the advanced stages, undergoing CCRT treatment, or receiving a radiation dose ≥60 Gy were significantly associated with CWL. However, these factors did not remain significant in the multivariate regression analysis. This suggests that these factors may have a negligible impact on CWL in patients with HNC.

Similar to what was reported by several studies [12,15,16,17,18], our study found that critical weight loss was more pronounced and independently associated with our study population who had a high pretreatment BMI. The majority of our study population had a higher pretreatment BMI: overweight (28%) and obese (26%). This finding contrasts with those of two studies that found no significant relationship between pretreatment BMI and critical weight loss among cancer patients with several types including HNC undergoing radiotherapy [6,19]. Furthermore, in recent research conducted by Ma et al. [15], they observed that HNC patients with a high BMI (overweight or obese) experienced significant weight loss. However, these patients were found to have more positive outcomes, such as improved treatment response, greater overall survival, and increased progression-free survival. On the other hand, HNC patients with a normal and underweight pretreatment BMI were at greater risk of poor overall survival [20,21,22,23], prognosis [24,25], and extended hospital stays for underweight patients [26].

With respect to the tumor site, HNC patients demonstrated different percentages of weight loss, which may affect optimal nutrition before and during treatment. The highest percentage of CWL was observed in patients with pharyngeal and oral cavity tumors, at 51% and 32%, respectively. Among pharyngeal tumor sites, 48% of nasopharyngeal cancer patients had a weight loss of ≥5% during treatment. Previous studies have shown that weight loss affects the nutritional status and risk of malnutrition adversely, particularly during the course of treatment [5,27]. In contrast, only 17% of patients with laryngeal, neck, and maxilla cancer experienced CWL. This result could reflect the presence of different symptoms, which vary in type and severity according to the disease location and treatment toxicities’ effect on nutritional intake. For example, dysphagia, mouth dryness, and taste changes are highly prevalent in nasopharyngeal and oral cancer [28,29]. These results highlight the importance of nutritional assessment and determining the tumor’s site prior to treatment, such as percutaneous endoscopic gastrostomy insertion, to minimize the intensity of the nutritional effect and ensure adequate nutrition [30].

In our study, sex was also significantly associated with CWL, as being male was found to be a risk factor around three times higher for CWL than being female. Of the 26 female patients included in the study, 24% developed CWL compared to 76% of the 119 male patients. However, previous studies have shown no significant difference in the effect of sex on CWL [12] and survival [31]. This result could be explained by the high prevalence of a high pretreatment BMI among men compared to women, which was found to be significantly associated with critical CWL during treatment.

Our study showed that patients who experienced CWL developed higher toxic side effects, mainly mucositis and mouth dryness, than non-CWL patients. This result is consistent with that of a previous report that showed high incidences of mucositis and mouth dryness in HNC patients undergoing radiotherapy with or without chemotherapy [32]. These toxic side effects induce oral pain and thus result in decreased oral intake and associated weight loss [33]. Therefore, to manage the impact of certain toxicities on weight loss, it is crucial to closely monitor and promptly treat the toxic side effects, mainly mucositis and mouth dryness, in HNC patients.

This study confirmed the relationship between the incidence of dry mouth and CWL (≥5%) during radiotherapy with or without chemotherapy. Furthermore, a significant correlation was found between CWL (≥5%) and the incidence of dry mouth. Higher rates of CWL (57%) in patients were found to be associated with the incidence of dry mouth compared to 31% of those with non-CWL. Similar to this result, other reports have shown that dry mouth is the most prominent complication by approximately 70% [34,35].

As our study was retrospective, the data available limited our ability to measure several other/different risk factors. Given the limited resources, nutritional status and interventions during treatment were difficult to collect and analyze. In addition, we were unable to obtain pre- and post-treatment weight loss and other adverse outcomes. Moreover, the BMI is affected by error due to an incorrect procedure for measuring the height when using a scale with a built-in altimeter according to the International Anthropometric Standardization reference manual [11]. Furthermore, the sample size was relatively small with the exclusion of 120 patients, which could have affected the results and the ability to generalize them. However, the study population’s homogeneity was considered the main strength of this study, as all patients underwent definitive radiotherapy with or without chemotherapy for HNC. In addition, all patients received the same radiotherapy technique, RapidArc^TM^, and oncologists assessed all side effects of radiotherapy with or without chemotherapy weekly for all patients included. Furthermore, all information was obtained from an electronic medical record designed for patients in the radiation field.

To the best of our knowledge, this is the first retrospective study to assess the ability to predict CWL in HNC patients undergoing radiotherapy with or without chemotherapy in Saudi Arabia. We recommend a larger sample size to further study the significant difference between pharyngeal tumor sites among different regions in Saudi Arabia. In addition, future research should attempt to investigate whether or not or to what extent different nutritional interventions related to adverse side effects could effectively prevent CWL during radiotherapy.

## 5. Conclusions

Weight loss is highly prevalent among HNC patients during treatment. Accordingly, high pretreatment BMI, cancer in the oral cavity, and being male could be considered predictive factors for CWL. However, factors such as having nasopharyngeal cancer, being in stage III or IV, undergoing CCRT treatment, or receiving a radiation dose of at least 60 Gy were found to have minimal impact on CWL. Mucositis and dry mouth are common toxic side effects of weight loss in HNC patients. It is important to closely monitor and treat these toxicities to minimize their impact on weight loss.

As a future direction, it is recommended to assess pretreatment BMI, percentage of weight loss, and nutritional intake and its influence on other parameters, e.g., body composition, in a prospective study design. A further recommendation would be to monitor treatment toxicities with respect to weight changes for each week during the treatment for a more defined correlation. In addition, research that involves recruiting a sample of patients receiving different types of chemotherapy and monitoring their nutritional status throughout their treatment is warranted. By comparing the nutritional status of patients receiving different types of chemotherapy, researchers can determine if certain treatments have a more significant impact on nutritional status than others.

## Figures and Tables

**Figure 1 cancers-16-00414-f001:**
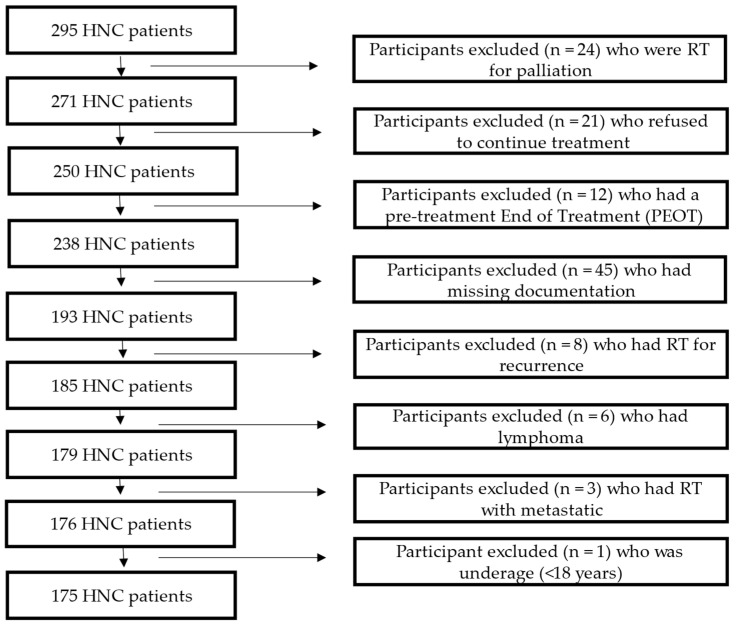
Sample distribution according to inclusion and exclusion criteria.

**Figure 2 cancers-16-00414-f002:**
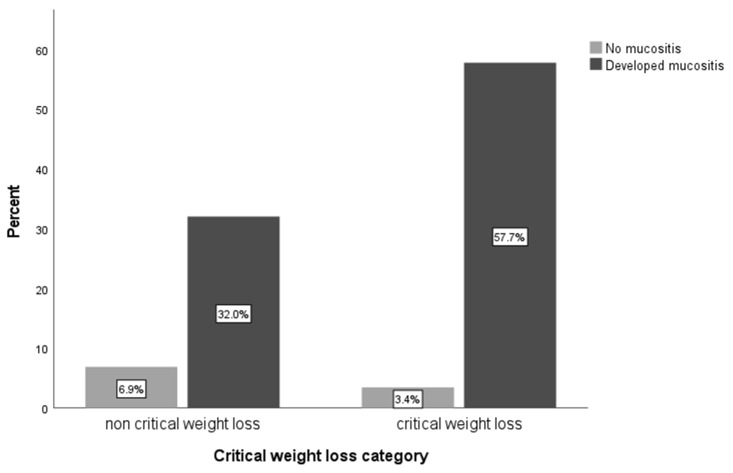
Relationship between mucositis incidence and CWL (OR = 3.6, 95% CI = [1.28–10.13], *p* = 0.02).

**Figure 3 cancers-16-00414-f003:**
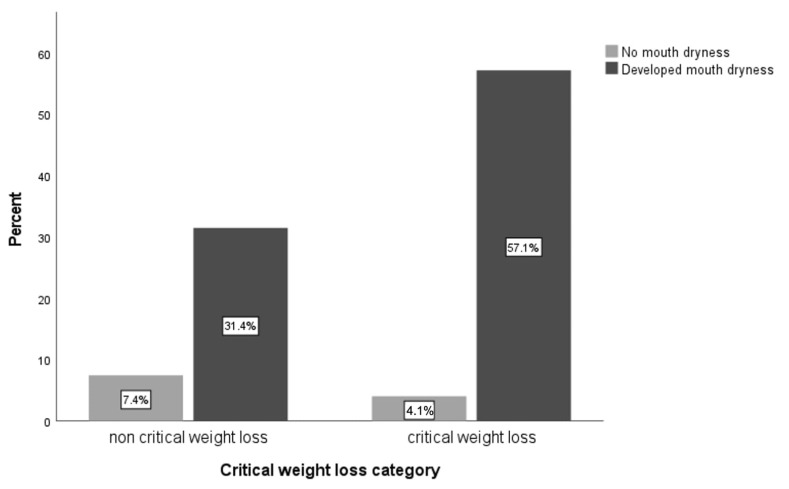
Relationship between mouth dryness incidence and CWL (OR = 3.37, 95% CI = [1.27–8.96], *p* = 0.02).

**Table 1 cancers-16-00414-t001:** Sample characteristics (*n* = 175).

Characteristics	*n* (%) or Mean ± SD
Sex	
Male	119 (68)
Female	56 (32)
Age (year)	54.6 ± 15.1
Height (cm)	163.6 ± 10.3
Pretreatment weight (kg)	70.8 ± 19.5
Pretreatment BMI (kg/m^2^)	26.2 ± 6.2
Post-treatment weight (kg)	65.7 ± 17.7
Post-treatment BMI (kg/m^2^)	24.4 ± 5.7
Weight loss (%)	6.7 ± 6
Critical weight loss	
Yes	107 (61)
No	68 (38)
Pretreatment BMI categories	
Underweight	16 (9)
Normal	64 (37)
Overweight	49 (28)
Obese	46 (26)

Abbreviations: SD: standard deviation, kg: kilogram.

**Table 2 cancers-16-00414-t002:** Characteristics of HNC patients based upon CWL.

Characteristics	CWL *n* (%) or Mean ± SD	Non-CWL *n* (%) or Mean ± SD	*p*-Value
Age (years)	53.2 ± 14.5	56.9 ± 15.9	0.12 †
Age category ≤65 years >65 years	89 (83) 18 (17)	51 (75) 17 (25)	0.13 ‡
Sex Male Female	81 (76) 26 (24)	38 (56) 30 (44)	**0.005** ‡
Diagnosis Oral cavity Pharynx Larynx Other (neck + maxilla)	34 (32) 55 (51) 17 (16) 1 (1)	32 (47) 14 (21) 13 (19) 9 (13)	**<0.001** ‡
Tumor site Nasopharynx Hypopharynx Oropharynx Non-pharyngeal	51 (48) 4 (3.1) 1 (0.9) 51 (48)	7 (10) 6 (9) 0 (0) 55 (81)	**<0.001** ‡
Disease stage (TNM) Early stage (Stages I and II) Advanced stage (Stages III and IV)	11 (10) 96 (90)	16 (23.5) 52 (76.5)	**0.02** ‡
History of surgery (tumor excision) No Yes	83 (78) 24 (22)	39 (57) 29 (43)	**0.004** ‡
Intention of treatment Without chemotherapy With chemotherapy	25 (23) 82 (77)	33 (48.5) 35 (51.5)	**0.001** ‡
Radiation dose <60 Gy ≥60 Gy	2 (2) 105 (98)	8 (12) 60 (88)	**0.008** ‡
Pretreatment weight (kg)	74.6 ± 20.4	64.8 ± 16.5	**0.001** †
Pretreatment BMI (kg/m^2^)	27.2 ± 6.6	24.7 ± 5.4	**0.01** †
Post-treatment weight (kg)	66.8 ± 19.0	63.9 ± 15.6	0.29 †
Post-treatment BMI (kg/m^2^)	24.4 ± 6.2	24.4 ± 5.1	0.98 †

Bold results are statistically significant at *p* < 0.05. ‡ Analysis performed with chi-square test, † Analysis performed with *t*-test. Abbreviations: BMI: body mass index, Gy: grey, SD: standard deviation, kg: kilogram.

**Table 3 cancers-16-00414-t003:** Regression analysis for the prediction of CWL in HNC patients.

Variables	OR (95% CI)	*p*-Value	AOR (95% CI)	*p*-Value
Sex Male Female	2.46 (1.28–4.71) 1	**0.007**	3.15 (1.39–7.11) 1	**0.006**
Diagnosis Oral cavity Pharynx Larynx Other (neck and maxilla)	9.56 (1.14–79.79) 35.35 (4.12–302.83) 11.76 (1.31–105) 1	**0.04** **0.001** **0.03**	10.36 (1.13–94.55) 4.99 (0.14–177.57) 8.58 (0.73–100.58) 1	**0.04** 0.38 0.09
Tumor site Nasopharynx Hypopharynx Oropharynx Non-pharyngeal	7.85 (3.26–18.88) 0.89 (0.25–3.12) ND 1	**<0.001** 0.87	9.43 (0.55–160.85) 2.18 (0.09–46.57) ND 1	0.12 0.63
Disease stage (TNM) Advanced stage (Stages III and IV) Early stage (Stages I and II)	2.68 (1.16–6.21) 1	**0.02**	0.2 (0.67–6.19) 1	0.21
History of surgery Yes No	0.38 (0.2–0.75) 1	**0.005**	1.03 (0.33–3.17) 1	0.96
Intention of treatment With chemotherapy Without chemotherapy	3.09 (1.60–5.94) 1	**<0.001**	10.36 (1.13–94.55) 1	0.12
Radiation dose ≥60 Gy <60 Gy	7 (1.43–34.04) 1	**0.02**	5.15 (0.60–44.12) 1	0.14
Pretreatment BMI (kg/m^2^)	0.93 (0.88–0.98)	**0.01**	1.10 (1.02–1.17)	**0.007**

The reference category is the “No CWL.” Bold results are statistically significant at *p* < 0.05. Abbreviations: AOR: adjusted odds ratio, OR: odds ratio, CI: confidence interval, kg: kilogram, Gy: grey, ND: not determined.

## Data Availability

The raw data supporting the conclusions of this article will be made available by the authors without undue reservation. Please contact Firas Azzeh (fsazzeh@uqu.edu.sa).

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
