# Peer review of "Predictive Factors for Critical Weight Loss in Saudi Head and Neck Cancer Patients Undergoing (Chemo)Radiotherapy"

_cancers, 2024, doi:10.3390/cancers16020414_

Round 1

Reviewer 1 Report

Comments and Suggestions for Authors

The limits of the study described in the manuscript should be indicated to improve the manuscript. If the body weight and height are not detected according to standardized procedures, all the work done has errors.

The nurse detected the patient's weight (the authors wrote this). How was body weight detected? Was the patient in underwear, or was he dressed? Is weighing a standardized procedure, or do the weighing methods depend on each individual nurse? I would ask the authors to describe better how body weight has been measured, given that their manuscript is based precisely on weight modification.

I ask the authors to write clearly within the study's limits that the height is to be measured with a stadiometer or a wall altimeter, as per the International Anthropometric Standardization reference manual. Using a scale with a built-in altimeter generates errors. It should, therefore, be written within the study's limits that the height (and therefore the BMI) is affected by error due to an incorrect procedure to detect the height.

I ask the authors to correct the following errors in the text of their manuscript:

Line 30: 25kg/m2 = 25 kg/m2

Line 108: I would compose Figure 1 differently. So, the way Figure 1 was drawn I think it creates confusion.  Rather than making a straight line, I suggest making two lines (a line for the included and a second line for the excluded), after which, only for the excluded do the line with the various reasons that caused the exclusion. 

Line 112: stages. Weight = In the pdf of the manuscript, there seems to be a double space after the point.

Line 120: ≥25-30 kg/m2 (overweight) = ≥25-29.9 kg/m2 (overweight)

Line 123: ≥ 60 = I ask the authors always to use the same format to report the information: after the symbol, there is a space in all the text written previously. There is no space after the symbol.

Line 141: P <0.05 = p <0.05 = The p value (or p-value) is written with the "p" in lowercase; the authors are asked to correct the "p" in all the text (it should not be written in capital letters).

Line 154: Table 1: BMI categories = Does the classification of patients according to BMI refer to pre-treatment? Please specify in the table. 

Line 188: gender (OR = In the pdf of the manuscript, there seems to be a double space after gender.

Author Response

Dear respected reviewer

Thank you for reviewing our paper entitled “Predictive Factors for Critical Weight Loss in Saudi Head and Neck Cancer Patients Undergoing (Chemo)Radiotherapy”. We have reviewed and corrected our paper based on your notes. The revised version of the manuscript is attached.

All reviewer comments in the revised manuscript have been changed and highlighted in yellow color. Concerning your comments, the following is our reply:

1) The limits of the study described in the manuscript should be indicated to improve the manuscript. If the body weight and height are not detected according to standardized procedures, all the work done has errors.

The answer:

We added the reference related to standardized procedures for weight and height, as the anthropometric measurements were determined according to the standard procedures in the hospital (lines# 115-116 in the revised manuscript, and reference 11 was added to the list of references).

The nurse detected the patient's weight (the authors wrote this). How was body weight detected? Was the patient in underwear, or was he dressed? Is weighing a standardized procedure, or do the weighing methods depend on each individual nurse? I would ask the authors to describe better how body weight has been measured, given that their manuscript is based precisely on weight modification.

The answer:

The body weight was measured for all participants according to the standard procedure, as were patients instructed to stand straight, wear light clothing, and be barefoot for all measurements (lines# 116-117 in the revised manuscript).

I ask the authors to write clearly within the study's limits that the height is to be measured with a stadiometer or a wall altimeter, as per the International Anthropometric Standardization reference manual. Using a scale with a built-in altimeter generates errors. It should, therefore, be written within the study's limits that the height (and therefore the BMI) is affected by error due to an incorrect procedure to detect the height.

The answer:

Added to the limitation section (lines# 282-285 in the revised manuscript

I ask the authors to correct the following errors in the text of their manuscript:

Line 30: 25kg/m2 = 25 kg/m2

The answer:

Changed (line# 30 in the revised manuscript)

Line 108: I would compose Figure 1 differently. So, the way Figure 1 was drawn I think it creates confusion.  Rather than making a straight line, I suggest making two lines (a line for the included and a second line for the excluded), after which, only for the excluded do the line with the various reasons that caused the exclusion. 

The answer:

Changed (line# 108 in the revised manuscript)

Line 112: stages. Weight = In the pdf of the manuscript, there seems to be a double space after the point.

The answer:

Changed (line# 111 in the revised manuscript)

Line 120: ≥25-30 kg/m2 (overweight) = ≥25-29.9 kg/m2 (overweight)

The answer:

Changed (line# 121 in the revised manuscript)

Line 123: ≥ 60 = I ask the authors always to use the same format to report the information: after the symbol, there is a space in all the text written previously. There is no space after the symbol.

The answer:

Changed in line# 125 in the revised manuscript. Also changed in all places in the revised manuscripts to be with a similar format.

Line 141: P <0.05 = p <0.05 = The p value (or p-value) is written with the "p" in lowercase; the authors are asked to correct the "p" in all the text (it should not be written in capital letters).

The answer:

It has been changed in all places in the revised manuscript.

Line 154: Table 1: BMI categories = Does the classification of patients according to BMI refer to pre-treatment? Please specify in the table. 

The answer:

Changed (line# 156 in the revised manuscript)

Line 188: gender (OR = In the pdf of the manuscript, there seems to be a double space after gender.

The answer:

Changed (line# 190 in the revised manuscript)

Finally, we want to thank you for your valuable comments and hope all corrections are performed correctly.

Sincerely Yours,

Dr. Firas Azzeh

Correspondence author

Reviewer 2 Report

Comments and Suggestions for Authors

This manuscript does not approach a very novel topic, as it is already known that chemo and radiotherapy can impact in the nutritional status and weight of patients, as well as their cancer.

However, the study gives specific data about the reach of this problem. 

I think that in relation of the chemotherapy, the authors should have specified the drugs used, especially if some of them can have more impact in the adverse effects that could lead to weight loss.

As mucositis and dry mouth are some of the problems related with weight loss, the authors could point out in the discussion or conclusions, the importance of preventing and treating these problems.

Author Response

Dear respected reviewer

Thank you for reviewing our paper entitled “Predictive Factors for Critical Weight Loss in Saudi Head and Neck Cancer Patients Undergoing (Chemo)Radiotherapy”. We have reviewed and corrected our paper based on your notes. The revised version of the manuscript is attached. 

All reviewer comments in the revised manuscript have been changed and highlighted in yellow color. Concerning your comments, the following is our reply:

This manuscript does not approach a very novel topic, as it is already known that chemo and radiotherapy can impact in the nutritional status and weight of patients, as well as their cancer. However, the study gives specific data about the reach of this problem. 

The answer:

The novelty of this paper is that it is the first study to assess the ability to predict CWL in HNC patients undergoing radiotherapy with or without chemotherapy in Saudi Arabia. This point is highlighted in the discussion section before the conclusion (line# 293 in the revised manuscript)

I think that in relation of the chemotherapy, the authors should have specified the drugs used, especially if some of them can have more impact on the adverse effects that could lead to weight loss.

The answer:

Thank you for your valuable comment. I understand your concern about the lack of information regarding specific chemotherapy drugs used in the paper, however, we think that this point could be considered in other studies since the retrospective nature of the study has some limitations such as the used drugs are not determined for all patients. Most of the patients used Cisplatin drug, but we can not generalize this point to all patients. Therefore, we highlighted this point to be in further studies as shown in lines# 312-316 in the revised manuscript.

As mucositis and dry mouth are some of the problems related with weight loss, the authors could point out in the discussion or conclusions, the importance of preventing and treating these problems.

The answer:

This excellent point has been added to the revised manuscript in the discussion (lines# 270-272), conclusion section (lines# 305-307), and recommendations after discussion (lines# 310-312).

Finally, we want to thank you for your valuable comments and hope all corrections are performed correctly.

Sincerely Yours,

Dr. Firas Azzeh

Correspondence author

Round 2

Reviewer 1 Report

Comments and Suggestions for Authors

The changes made to the manuscript by the authors allow readers to interpret the data in the manuscript better. 

I give the reference of the manual of anthropometric standardization.

https://archive.org/details/anthropometricst0000unse/mode/2up